# Random RotBoost: An Ensemble Classification Method Based on Rotation Forest and AdaBoost in Random Subsets and Its Application to Clinical Decision Support

**DOI:** 10.3390/e24050617

**Published:** 2022-04-28

**Authors:** Shin-Jye Lee, Ching-Hsun Tseng, Hui-Yu Yang, Xin Jin, Qian Jiang, Bin Pu, Wei-Huan Hu, Duen-Ren Liu, Yang Huang, Na Zhao

**Affiliations:** 1Institute of Management of Technology, National Yang Ming Chiao Tung University, Hsinchu 300, Taiwan; camhero@gmail.com (S.-J.L.); eydie1995@gmail.com (H.-Y.Y.); 2Department of Computer Science, The University of Manchester, Manchester M13 9PL, UK; hank131415go61@gmail.com; 3National Pilot School of Software, Yunnan University, Kunming 650504, China; xinjin@ynu.edu.cn (X.J.); jiangqian@ynu.edu.cn (Q.J.); 4College of Computer Science and Electronic Engineering, Hunan University, Changsha 410082, China; pubin@hnu.edu.cn; 5College of Computer Science, National Yang Ming Chiao Tung University, Hsinchu 300, Taiwan; indi6748@gmail.com; 6Institute of Information Management, National Yang Ming Chiao Tung University, Hsinchu 300, Taiwan; dliu@mail.nctu.edu.tw (D.-R.L.); bokxko1023@gmail.com (Y.H.)

**Keywords:** classification, Rotation Forest, AdaBoost, clinical decision support

## Abstract

In the era of bathing in big data, it is common to see enormous amounts of data generated daily. As for the medical industry, not only could we collect a large amount of data, but also see each data set with a great number of features. When the number of features is ramping up, a common dilemma is adding computational cost during inferring. To address this concern, the data rotational method by PCA in tree-based methods shows a path. This work tries to enhance this path by proposing an ensemble classification method with an AdaBoost mechanism in random, automatically generating rotation subsets termed Random RotBoost. The random rotation process has replaced the manual pre-defined number of subset features (free pre-defined process). Therefore, with the ensemble of the multiple AdaBoost-based classifier, overfitting problems can be avoided, thus reinforcing the robustness. In our experiments with real-world medical data sets, Random RotBoost reaches better classification performance when compared with existing methods. Thus, with the help from our proposed method, the quality of clinical decisions can potentially be enhanced and supported in medical tasks.

## 1. Introduction

According to Mitchell, T., “Machine learning is the study of computer algorithms that improves automatically through experience.” [1]. That is to say, the collected data are classified or predicted by the machine learning algorithm. As the new data are obtained, the trained model can be applied for the purpose of prediction or classification, and then applied on a range of real-world fields. One of the popular fields is the medical industry, which generates enormous amounts of data with multiple features every day. When applying a machine learning algorithm on these types of data sets, increased data numbers and the expanding dimensions of features can create an observable increase in noise and time loss. This is especially true when ensemble decision trees are applied in the medical field. More trees [2,3,4] only increase the dilemma and cloud the prediction performance. The dilemma keeps impeding professional staff from making the right decision fast.

In order to support medical decision making and keep high time cost to a minimum, one of the key aspects of this work is the Rotation Forest method, which belongs to the forest-type algorithm of the decision tree family. It has been developed in many application scenarios with excellent performances, such as hyperspectral image classification [5], identification of cell pathology [6], etc. Meanwhile, as attention on the robustness from using trees increases, one of the advantages of decision trees is that they are explainable by viewing classifying value in each tree. Another advantage is that they are able of handling various types of data. By building a Rotation Forest (RotForest) [7] based on decision trees, it not only has the aforementioned advantages, but it could also decrease expensive time cost by using PCA to reduce features. In the rotation, the features are divided into K non-overlapping subsets of equal size, and then the integrating PCA [8] rotates subsets to new subsets with pre-defined fewer features. Latterly, rotation matrix and voting scene with bagging method have involved prediction. Although there have been many works on decision trees and Rotation Forest, such as isomerous multiple classifier [9], the shortcomings still include complicated calculations and high computation cost. Particularly, RotForest requires pre-knowledge to manually pre-define the K. Therefore, this work aims to develop a RotForest-based ensemble learning algorithm without pre-defining the K and rotated features. Namely, a random K is implemented in the proposed method and randomly rotated based on the random segment K data sets. Moreover, because most of the existing methods do not connect each rotated classifier, which indicates that each classifier is independent, this work builds the connection among classifiers using AdaBoost [10]. Hence, we called the proposed method Random Rotation Boost (Random RotBoost), which inherits advantages from ensemble decision trees and RotForest with stronger robustness from the proposed random process and connection. In our simulations, Random RotBoost outshines other ensemble learning methods in terms of accuracy, recall, precision, F1-score, and speed.

The process of this research work is organized as follows. Section 1 elaborates on the research background, purpose, and motivation. Section 2 discusses the relevant literature and the background knowledge, which laid the foundation of the proposed model. Next, the proposed method and comparisons with others are presented in Section 3 and Section 4, respectively. Section 5 draws conclusions and presents future works, along with the extra materials toward this work in Appendix A.

## 2. Related Works

In this section, the literature reviews of ensemble learning, PCA transformation, and the Rotation Forest are discussed. The narrative in this section helps to understand the background knowledge for the proposed model.

### 2.1. Decision Trees

With the gradual expansion of machine learning, a variety of machine learning algorithms have been developed with niche functions, such as decision tree. The decision tree is a branch with long history and diverse kinds of machine learning algorithms. Its learning strategy is based on the reasoning principle of loss function minimization, which is conducted via a top-down recursive mechanism to calculate the information contribution of each feature type. There are various decision tree algorithms, such as classification and regression trees (CART) [2], ID3 [3], and C4.5 [4]. By using this information to split the category of data, we have witnessed not only applications on classification [11], but also on support decision with hypotheses [12,13,14].These tree algorithms have built the foundation and the application examples for the following ensemble learnings and inspired this work.

### 2.2. Ensemble Learning

Ensemble learning [15] is a supervised learning algorithm proposed by Opitz, D. and Maclin, R., which applies multiple learning models to enhance the prediction performance. Its comprehensive performance is usually stronger than that of any single model, so it usually has better performance than a separate learning method. Among the ensemble learnings, the Forest is one of the most widely known and combines weak learners (tree) to form a more accurate and stronger learner. According to the sampling method, strong learners can further be divided into two main categories: bagging [16] and boosting [17]. The core idea of bagging is randomly sampling and then voting on the training sample set. A typical representation of this is Random Forest [18]. In addition, the mechanism of boosting aims to adjust weight in the sampling, which stands for GBDT [19], XGBoost [20], etc. These algorithms have won numerous awards in the Kaggle contest in recent years and have excellent performance in many application fields [21]. However, evaluating the performance of ensemble learning models usually requires more computation cost than a single learning model.

### 2.3. Adaptive Boosting (AdaBoost)

The AdaBoost algorithm is also a boosting ensemble learning method [22]. It is a popular method for improving quality, and it constructs strong classifiers through linear combination of weak classifiers. The boosting decision tree algorithm is a boosting method using the classification and regression trees (CART) as the base classifier. It can be stated that the decision tree algorithm in this situation is a special case of AdaBoost. As the linear combination of decision trees can effectively fit the training data, the boosting decision tree is also a well-performing learning algorithm [10]. During the training process, each weak classifier is trained in turn, and their weights are obtained, and the previously misclassified samples are dynamically adjusted. Samples that are misclassified by the previous weak classifiers will increase the weight, and the opposite will reduce the weight. The purpose of updating the weight is to improve the most useful samples in each successive iteration. Finally, the prediction result of the final classifier is the weighted sum of the prediction results generated by the weak classifiers. However, what needs to be overcome is that it is sensitive to noise data, which has been discussed as a common issue for decision trees [21,23]. Toward this sensitivity to noise, rotation operation by PCA could be the cure.

### 2.4. Principal Components Analysis (PCA)

PCA was proposed by Pearson, K. [8] for analyzing data and establishing mathematical models, also known as Karhunen–Loeve Transform. It is applied in machine learning as a technique for dimensionality reduction, compression, and simplification of data sets. Additionally, it can be applied for the exploration and visualization of high-dimensional data. The main steps aim to normalize the data and then perform singular value (SVD) [24] feature decomposition on the covariance matrix to obtain the eigenvector, eigenvalue of the data, and the feature values in descending order. During the process, K eigenvalues and feature vectors are selected and projected into a low-dimensional subspace to achieve dimensionality reduction for new features. The new low-dimensional data set preserves the variables of the original data as much as possible. The dimensions changing operation of PCA is also applied to retain the most important and valuable features in the data set, so the overall performance will not degrade. Moreover, a better result could be possible because the projected data set is simplified by reducing the concern of overfitting. It is quite practical at analyzing complex data. However, the interpretability is highly impacted by pre-defined K, so the needed prior-knowledge could dim the merit of PCA during the process.

### 2.5. Rotation Ensemble Learning

As per the ensemble learning discussion in Section 2.1, Forest-type can be divided into bagging and boosting. In the light of Rotation, we can see a similar picture as Forest: Rotation Forest (RotForest) for bagging, Rotation Boosting (RotBoost) for boosting.

Rotation Forest [7] is an ensemble learning method based on feature extraction by PCA. The rotation operation aims to ensure the performance of the classifier while generating different training subsets to enhance the diversity. Thus, with the same process of voting in prediction as Forest, it has better prediction performance. Apart from tabular data classification, RotForest is also active in classifying images [25,26]. Compared with the support vector machine-recursive feature elimination (SVM-RFE), RotForest shows a superior performance for all subsets. In addition, the integration of RotForest and AdaBoost, which is using AdaBoosting as the final voting part, is applied for hyperspectral image classification [5]. The integration follows RotBoost [27], whose base classifiers are decision trees with AdaBoost to improve the performance by adjusting the error. Most importantly, it operates in a parallel way, so the reduction of error toward each base classifier is applied after calculating the loss as a whole. The operation shares a similar aspect of this work. We elaborate on this in Section 3.

## 3. Random Rotation Boosting

Based on PCA, AdaBoost, and ensemble learnings, the detail of the proposed work Random RotBoost is as follows. Before the detailed explanation, a series of primaries toward our proposed method is revealed in advance, including RotForest, AdaBoost, and RotBoost.

### 3.1. Prelimanries and Notations

Throughout this work, we set a binary classification training data set as L, where $L=\left\{\left(x_{1\;},\;y_{1}\right),\;\left(x_{2\;,}\;y_{2}\right),\;\cdots ,\;\left(x_{N\;,}\;y_{N}\right)\right\}$, $x_{i}\in X\subseteq R^{N}$, $y_{i}=\left\{-1,+1\right\}$, number of data as N, number of features as F, number of subsets as K, initial weight distribution as $D,\;where\;D_{i}=\left(w_{1n},\cdots w_{1N}\right)$, iteration as i, sample as S, classifiers as t.

#### 3.1.1. RotForest

In RotForest, for the initial S, F is randomly divided into K subsets (can be defined according to the requirement), and these subsets may be intersecting or disjointed (disjoint scenario is applied in this work). After obtaining K, each feature subset has M features, M=F/K. If the feature number cannot be divisible, the remaining features are added to the last set of features. The division of feature subsets determines the new samples generated, which affects the differences among the ensemble classifiers. Then, K samples are resampled to obtain a bootstrap sample, and PCA is applied to the sample subset of classes. PCA stores the principal component coefficients $a_{v,j}^{\left(1\right)},a_{v,j}^{\left(2\right)},\cdots ,a_{v,j}^{\left(M\right)}$ to preserve the important parts of the data and variability information. After performing PCA transformation processing on all K sample subsets, the K-axis rotation of the data set occurs, and all the coefficients are combined to generate a sparse rotation matrix $R_{i}$ that can be presented as follows:(1)$$R_{i}=\left[\begin{array}{ccc} a_{v,j}^{\left(1\right)},a_{v,j}^{\left(2\right)},\cdots ,a_{v,j}^{\left(M_{1}\right)} & \cdots & \left[0\right]\\ \vdots & \ddots & \vdots \\ \left[0\right] & \cdots & a_{v,j}^{\left(1\right)},a_{v,j}^{\left(2\right)},\cdots ,a_{v,j}^{\left(M_{K}\right)} \end{array}\right]$$
where $R_{i}$ is rearranged according to the order of the original features to get $R_{i}^{a}$, so the new training set of the classifier is $XR_{i}^{a}$. By getting the new training set, the whole process of RotForest [7] is below.

**Input:**L, loss: l.

**For** t=1,2,⋯,T:

Split the feature set *F* into *K* subsets; each subset has M=F/K features;Draw a bootstrap sample of every subset and apply PCA to get the principal component coefficients;Repeat the step 1 and 2 K times and put the K principal component coefficients into the rotation matrix;Rearrange the rotation matrix according to the order of the original feature set, and then the training set of the classifier $G_{t}\left(x\right)$ is $XR_{t}^{a}$;Get the classification result of $G_{t}\left(x\right)$.


**Output:**


The final classifier is as follows:(2)$$G\left(x\right)=\frac{1}{M}\overset{T}{\underset{t=1}{\sum }}G_{t}\left(x\right)$$

The above process can also be shown as Figure 1.

#### 3.1.2. AdaBoost

In our proposed method, one of the important parts is to connect each rotation result via AdaBoost [10]. The following process shows the typical detailed steps of AdaBoost.

**Input:**L, loss: l, initial weight D

**For** t=1,2,⋯,T:

Calculate the classifier error of $G_{t}\left(x\right)$ from l on training set:(3)$$e_{t}=P\left(\left(G_{t}\left(x_{i}\right)\neq y_{i}\right)\right)=\overset{N}{\underset{i=1}{\sum }}w_{ti}l\left(G_{t}\left(x_{i}\right)\neq y_{i}\right)$$Calculate the classification of $G_{t}\left(x\right)$:(4)$$\alpha_{t}=\frac{1}{2}\log\frac{1-e_{t}}{e_{t}}$$Update the weight distribution:(5)$$D_{t+1}=\left(w_{t+1,n},\;\cdots ,w_{t+1,N)}\right)$$Build a linear combination:(6)$$f\left(x\right)=\overset{M}{\underset{t=1}{\sum }}\alpha_{t}G_{t}\left(x\right)$$
where $w_{t+1,\;n}=\frac{w_{tn}}{z_{t}}\exp\left(-\alpha_{t}y_{n}G_{t}\left(x\right)\right)$ and $Z_{t}$ is a normalization factor, so that $D_{t+1}$ is a probability distribution.


**Output:**


The final classifier is as follows:(7)$$G\left(x\right)=sign\left(f\left(x\right)\right)=sign\left(\overset{T}{\underset{t=1}{\sum }}\alpha_{t}G_{t}\left(x\right)\right)$$

#### 3.1.3. RotBoost

In Section 2, RotBoost [27] shares a similar aspect of this by simply combining RotBoost with AdaBoost. In order to train multiple base classifiers parallelly, AdaBoost minimizes the residual from the aggregation of all classifier results. Based on the elaborations of RotForest and AdaBoost in the last two sub-sections, the process of RotBoost can be represented in the following description along with Figure 2:

Apply the same steps of 1 to 4 from RotForest to obtain the training data;Apply the decision tree as the base classifier;Update the weights of each classifier based on the aggregation result of all classifiers, which is following the steps of 1 to 3 by calculating the residual from the output of G(x) in (7).

### 3.2. Random RotBoost

In order to take the advantages from RotForest and AdaBoost and further improve them, a pre-defined free random Rotation with adaptive boosting connection Forest is Proposed entitled Random RotBoost. The training process is different from the process detailed above with AdaBoost connecting each tree between each rotation, and each rotation part conducting randomly featured projections on random segment subsets before processing PCA conversion. That is, instead of predefining *K* as a number of rotation sample subsets, a random number is applied for randomly selecting feature subsets at each round, and this design increases the diversity of features. After processing the PCA conversion, the useful information of features can be retained to the greatest extent, and the redundant information, as well as less impactful features, can also be effectively removed. The following process with Figure 3 explains the whole process of Random RotBoost.

**Input:**L, loss: l, initial weight D

For *f* in *F*:

Randomly select K:
(a)Split the feature set *F* into *K* subsets; each subset has M=F/K features;(b)Process PCA to get a rotation matrix;(c)Apply decision tree to perform the classification and calculate the weight of each time.

AdaBoost connect:
(a)Calculate the classifier error of $g_{f}\left(XR_{f}^{a}\right)$ from l on the training set:(8)$$g_{f}=P\left(\left(g_{f}\left(x_{i}\right)\neq y_{i}\right)\right)=\overset{N}{\underset{i=1}{\sum }}w_{fi}l\left(g_{f}\left(x_{i}\right)\neq y_{i}\right)$$(b)Calculate the classification of $g_{f}\left(x\right)$:(9)$$\alpha_{f}=\frac{1}{2}\log\frac{1-e_{f}}{e_{f}}$$(c)Update the weight distribution:(10)$$D_{f+1}=\left(w_{f+1,n},\;\cdots ,w_{f+1,N)}\right)$$(d)Build the updated combination:(11)$$g_{f}\left(x\right)=sign\left(\overset{f}{\underset{i=1}{\sum }}\alpha_{i}G_{i}\left(x\right)\right)$$


**Output:**


The final classifier is as follows:(12)$$G\left(x\right)=sign\left(f\left(x\right)\right)=sign\left(\overset{F}{\underset{f=1}{\sum }}\alpha_{f}G_{f}\left(x\right)\right)$$

## 4. Experiments and Results

### 4.1. Data Sets and Evaluation Metric

In our simulations, nine data sets are selected from UCI ML Repository [28] for evaluating the reliability of the proposed method and the diversity of simulations. These data sets involve different fields, including medicine, machinery, etc. Especially in the medical field, there are 6 medical data sets, including Brainwave, Parkinson, Breast-Cancer, Wisconsin-Breast, Heart-Stalog, and real-world electrocardiogram (ECG) data. The data size and number of features of all data sets are summarized in Table 1.

The confusion matrix is applied to evaluate the classification performance of the proposed method. In our experiment, evaluation, accuracy, precision, recall, and F-measure (F1-score) [28] are applied. Accordingly, the following results and discussion are divided toward each index to discuss. Finally, a time cost observation toward each candidate is also revealed after the metrics evaluation.

### 4.2. Results and Discussion

The performance of Random RotBoost is compared with four ensemble methods, including the RotForest, RotBoost, Ensemble 1, and Ensemble 2. The detailed information about Ensemble 1 and Ensemble 2 can be seen in the Appendix A. Both ensembles follow the structure of RotForest, but each ensemble replaced the base classifier from the decision tree with a set of different classifiers. For example, Ensemble 1: decision tree, boosting tree, and Random Forest; Ensemble 2: Logistic Regression, boosting tree, and KNN. For each model with every data set, 80% of randomly sampled data are assigned as the training set, and 20% are assigned for testing accuracy, precision, recall and F1-score.

#### 4.2.1. Accuracy

Accuracy is one of the most intuitive ways of observing performance. In Figure 4, except for the performance in Parkinson, Random RotBoost outshines others in most of the medical data sets. In light of many difficult tasks in Truck, which consists of over 100 features, Random RotBoost dominates over competitors. On the other hand, although Breast-Cancer and Xdt20w(ECG) have non-obvious differences between features, Random RotBoost still maintains a better result.

#### 4.2.2. Precision

A similar picture in accuracy is witnessed in this precision comparison in Figure 5. By using precision, it reflects the ability of the model to distinguish between negative samples. Based on this expectation, Random RotBoost performed better than others in the Breast-Cancer data set, which could indicate our method is less likely to misdiagnose cancer-free patients. Moreover, the gap between Random RotBoost shrinks in terms of Parkinson, which further demonstrates Random RotBoost’s ability.

#### 4.2.3. Recall

When evaluating models with Recall in Figure 6, the assessment of models’ completeness and ability to classify positive results is considered In the medical industry, this type of capability is vital to recognize disease. We can see that Random RotBoost still has a high performance across the data sets. Although the results of Truck have degraded slightly, the gap between Parkinson and others has also degraded.

#### 4.2.4. F1-Score

After evaluating accuracy, precision, and recall, an all-around evaluating aspect is F1-Score, which is an index balancing the ability of evaluation of positive and negative capability. As our proposed method, Random RotBoost, keeps maintaining the high-level performance compared with others, we see a similar picture in the former three metrics. However, the result of RotBoost shows a much different picture in the former evaluations, which are quite different from metric to metric. Further, another surprising aspect is the unstable results of Ensemble 2, which has fluctuating results in different data sets. This could indicate that a combination of very diverse base classifiers will dim each other. Overall, from these evaluations, Random RotBoost maintains the merit of providing high-dimensional data to reflect on every metric performance, please see Figure 7.

#### 4.2.5. Time Cost

Apart from classifying performance, evaluating total prediction time is also an important aspect. In Section 3, the detailed mechanism of Random RotBoost was revealed. Adding the random rotation process and AdaBoost connection operations could increase time spent. Although a typical method is measuring the time complexity [23], as the proposed algorithm involves a random process, we conducted a comparison in average time spending in prediction. Therefore, a total time cost result in prediction is recorded in Table 2, where bold text means the best performance among the comparison. By directly comparing average costing time, Random RotBoost did not sacrifice too much time cost performance, which ranks second among all candidates. Most importantly, it costs the least time in Wisconsin-breast. When focusing on medical data sets, although the computation time of proposed method is not necessarily the shortest, it is not far from those of other ensemble methods. This demonstrates the superior performance of Random RotBoost toward medical data sets.

## 5. Conclusions and Future Works

### 5.1. Conclusions

This work effectively integrates the advantages of Rotation Forest and AdaBoost to propose a new tree-based algorithm, Random RotBoost. Regarding the concern of noise from sparse features in medical data sets, a less-solid classification method is not only unable to reach a good classification performance but can also easily cause the problem of data distortion in processing medical data. From the practical evidence, this work shows that the proposed method can reach a reliable classification performance in simulation experiments when compared with other novel ensemble methods. Thus, the proposed method can potentially enhance the quality of clinical decision support based on its prediction performance of processing medical data sets.

### 5.2. Future Works

In the future, we expect that the proposed method can be used in practical applications. When selecting experimental data sets, there are special types of medical fields, such as Brainwave, Parkinson, and Xdt20w (ECG). It can be said that the proposed method can help medical personnel to make good diagnostic predictions and support informed clinical decisions. Although Random RotBoost has improved the classification accuracy compared with traditional classification methods, there is still much room for improvement in the processing of small sample data sets. Determining how to deal with this type of data is still a major challenge in the field of machine learning.

## Figures and Tables

**Figure 1 entropy-24-00617-f001:**
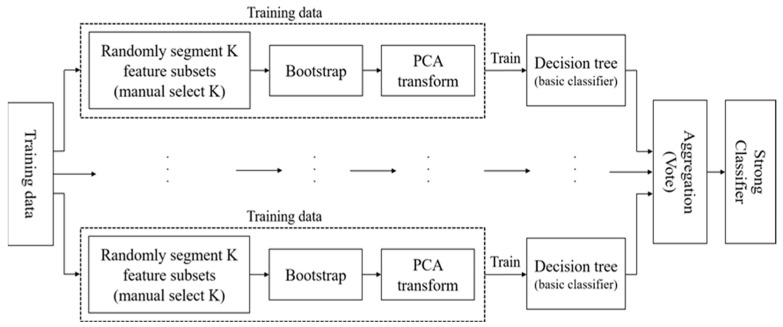
The Flowchart of RotForest.

**Figure 2 entropy-24-00617-f002:**
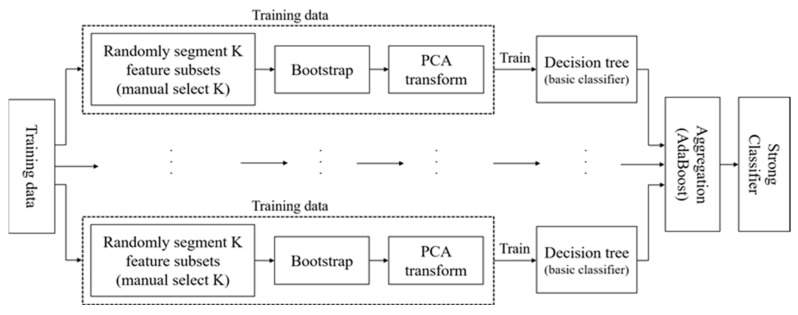
The Flowchart of RotBoost.

**Figure 3 entropy-24-00617-f003:**
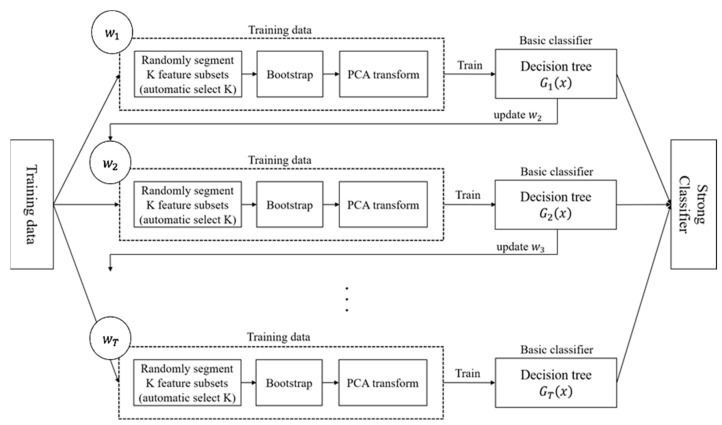
The flowchart of the proposed algorithm—Random RotBoost.

**Figure 4 entropy-24-00617-f004:**
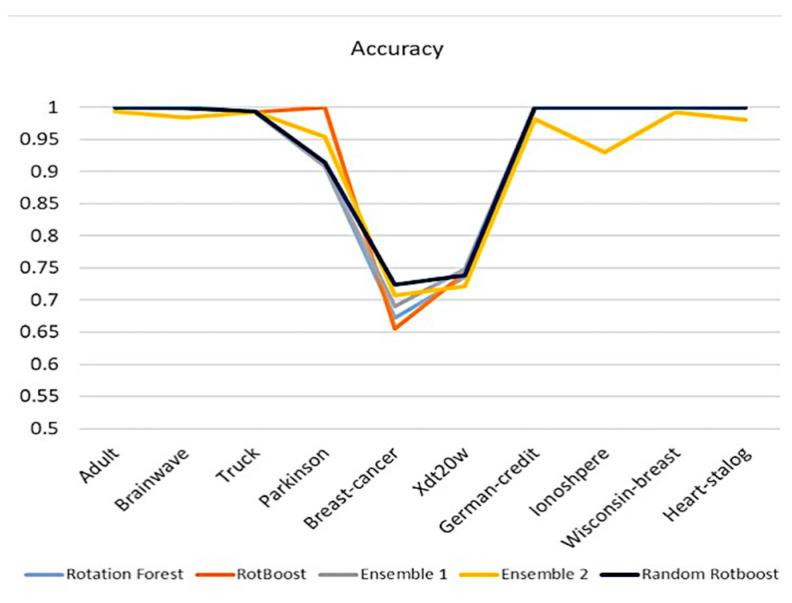
Accuracy Comparison.

**Figure 5 entropy-24-00617-f005:**
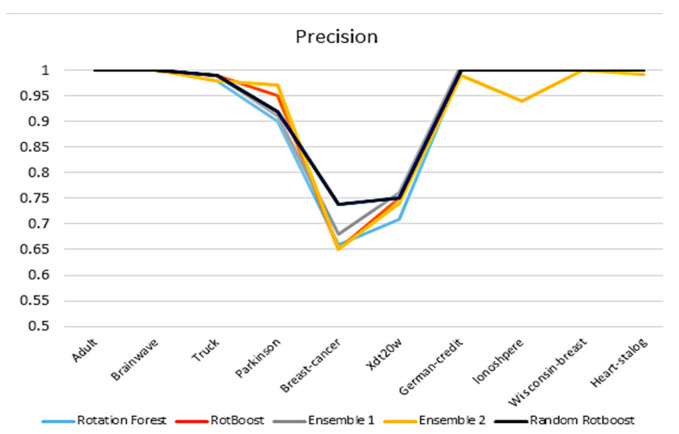
Precision comparison.

**Figure 6 entropy-24-00617-f006:**
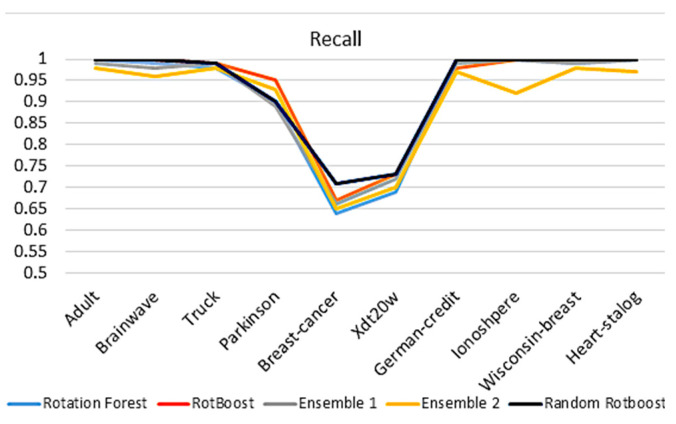
Recall comparison.

**Figure 7 entropy-24-00617-f007:**
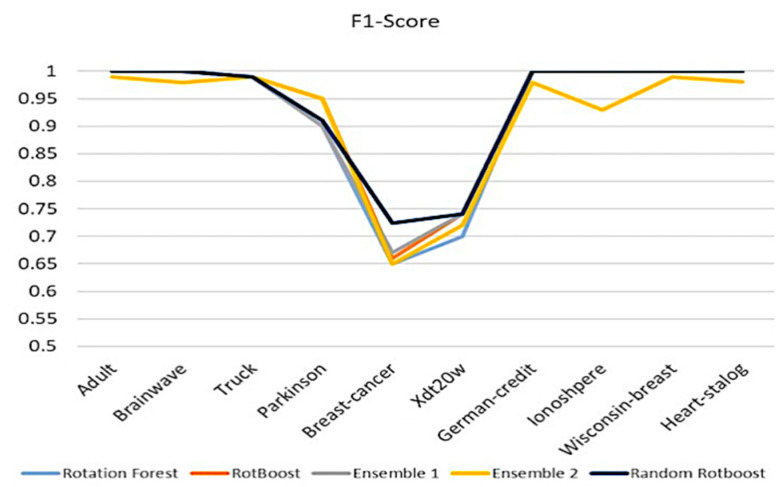
F1-Score comparison.

**Table 1 entropy-24-00617-t001:** Experimental data description.

Data Set	Instances	Attributes
Adult	48,842	14
Brainwave	12,811	15
Truck	60,000	171
Parkinson	197	23
Breast-cancer	569	32
Xdt20w(ECG)	202,594	14
German-credit	1000	20
Ionosphere	351	34
Wisconsin-breast	699	10
Heart-Stalog	270	13

**Table 2 entropy-24-00617-t002:** Total time cost comparison (time/s).

	Method	RotForest	RotBoost	Ensemble 1	Ensemble 2	Random RotBoost
Data Set						
Adult		**51.601**	85.179	2048.000	2525.200	174.665
Brainwave		51.601	**20.662**	144.048	83.250	127.250
Truck		6775.965	2394.330	10,078.85	**1592.149**	2103.997
Parkinson		324.470	**78.705**	1099.380	795.991	478.519
Breast-cancer		**100.952**	103.477	107.026	102.880	113.555
Xdt20w (ECG)		**220.200**	252.553	454.574	618.849	220.288
German-credit		100.780	**100.535**	105.992	100.659	102.749
Ionosphere		101.797	**100.820**	112.111	101.215	101.049
Wisconsin-breast		101.149	100.820	104.431	100.839	**100.741**
Heart-Stalog		**100.489**	100.494	105.336	100.679	100.818
Adult		**51.601**	85.179	2048.000	2525.200	174.665
Average		725.510	**311.159**	1491.613	786.083	345.300

## Data Availability

The simulation data can be accessed from UCI ML Repository: https://ergodicity.net/2013/07/ (accessed on 23 May 2019).

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
