# Peer review of "Random RotBoost: An Ensemble Classification Method Based on Rotation Forest and AdaBoost in Random Subsets and Its Application to Clinical Decision Support"

_entropy, 2022, doi:10.3390/e24050617_

Round 1

Reviewer 1 Report

Be placed in the article presented at Entropy a 2-3 Sources of classical works of the subject.

Author Response

  1. Several referred works at Entropy have been added, please see reference 11, 12, 13, 14, 21, 23, and 25.

Reviewer 2 Report

In this paper, an ensemble classification method based on rotation forest and Adaboost in random sub-sets has been proposed. The structure of this paper is well organized, and promising results are presented. Here are some suggestions for improving the quality of this paper:

  • In paragraph #1 of the Introduction, the description of the definition of machine learning can be simplified.
  • The development of the decision tree algorithms has not been well introduced. One reference [14] was included. It is insufficient. More literature review should be conducted. And also, the research gap should be well introduced.
  • As for the journal paper, ‘Section’ is widely used, instead of ‘Chapter’.
  • It is suggested to include a diagram to introduce the overall structure of the developed method.
  • Figure 1-20 is in low resolution.
  • Some discussion on the comparison results should be included.
  • It is suggested to have a recommendation for future work in the Conclusion.

Author Response

  1. The description about machine learning definition has been simplified, please see line 30-41.
  2. The research gap has been added in Section I from line 42-183. Also, the pros and cons of existing tree-based methods have been discussed in 2.1 of Section II, along with some variant works in the same section.
  3. In line 184-189, the used words of “Chapter” have been replaced with “Section”.
  4. The workflow of proposed method – Random RotBoost has been well explained in 3.2, along with flowchart in Figure 3.
  5. Figure 1 – 19 has been replaced with high resolution ones.
  6. The comparison candidates, RotForest, RotBoost, variant Ensemble 1 and 2, have been added the detail information in 3.1 as Preliminaries. Please see the detail flowchart of Ensemble 1 and 2 in Appendix 6.1.1 and 6.1.2.
  7. The conclusion has been refined with added future works toward this work.

Reviewer 3 Report

The manuscript has presented a Random Rotboost classification method based on Rotation Forest and AdaBoost for clinical datasets. The manuscript is well written along with well-discussed case studies for different types of clinical datasets. The results are impressive, however, the authors may consider the following comments to revise the manuscript:   1. The quality of the figures is very poor. It is advised to revise these figures with better resolutions and aesthetics.   2. The language of the manuscript is not clear in several places. It is advised to proofread the manuscript.   3. The cross-validation of the proposed method can add value to the case study. The author may consider the following article for better understanding.  https://www.mdpi.com/1996-1073/13/10/2578

Author Response

  1. Figure 1 – 19 has been replaced with high resolution ones.
  2. The English throughout the paper has been revised.
  3. As the whole revision needs to be done within five days, after thoughtfully consideration, we are afraid that the cross-validation part toward a cased study will not be conducted in our work. However, we are grateful your kindly suggestion.